



# Observations and Simulations of a Wind Farm Modifying a Thunderstorm Outflow Boundary

Jessica M. Tomaszewski[1] and Julie K. Lundquist[1,2]

[1]Department of Atmospheric and Oceanic Sciences, University of Colorado, Boulder, CO 80309-0311, USA
[2]National Wind Technology Center, National Renewable Energy Laboratory, Golden, CO 80401-3305, USA

**Correspondence:** Jessica M. Tomaszewski (jessica.tomaszewski@colorado.edu)

**Abstract.** On June 18, 2019, National Weather Service (NWS) radar reflectivity data indicated the presence of thunderstorm-generated outflow propagating east-southeast near Lubbock, Texas. A section of the outflow boundary encountered a wind farm, and then experienced a notable reduction in propagating speed, suggesting that interactions with the wind farm impacted the outflow boundary progression. We use the Weather Research and Forecasting model and its Wind Farm Parameterization

to address the extent to which wind farms can modify thunderstorm outflow boundaries. We conduct two simulations of the June 2019 outflow event, one containing the wind farm and one without. We specifically investigate the outflow propagation speed of the section of the boundary that encounters the wind farm and the associated impacts to near-surface wind speed, moisture, temperature, and changes to precipitation features as the storm and associated outflow pass over the wind farm domain. The NWS radar and nearby West Texas Mesonet surface stations provide observations for validation of the simulations.

The presence of the wind farm in the simulation clearly slows the progress of the outflow boundary by over $20\,\mathrm{km\,hr^{-1}}$, similar to what was observed. Simulated perturbations of surface wind speed, temperature, and moisture associated with outflow passage were delayed by up to 6 minutes when the wind farm was present in the simulation compared to the simulation without the wind farm. However, impacts to precipitation were localized and transient, with no change to total accumulation across the domain.

*Copyright statement.* This work was authored [in part] by the National Renewable Energy Laboratory, operated by Alliance for Sustainable Energy, LLC, for the U.S. Department of Energy (DOE) under Contract No. DE-AC36-08GO28308. Funding provided by the U.S. Department of Energy Office of Energy Efficiency and Renewable Energy Wind Energy Technologies Office. The views expressed in the article do not necessarily represent the views of the DOE or the U.S. Government. The U.S. Government retains and the publisher, by accepting the article for publication, acknowledges that the U.S. Government retains a nonexclusive, paid-up, irrevocable, worldwide license to publish or

reproduce the published form of this work, or allow others to do so, for U.S. Government purposes.

## 1 Introduction

Wind energy deployment is growing rapidly to provide a near-zero emissions source of electricity that can meet increasing energy demands. The International Energy Agency (IEA) predicts wind energy will reach 14% of global capacity ($\sim 1,700$



GW) by 2040 (IEA, 2018). Wind turbines generate electricity by using momentum from the wind to turn their blades and
generator, causing a downwind wake characterized by an increase in turbulence and reduction in wind speed (Lissaman,
1979). Groups of turbines will then generate an aggregate wind farm wake, which has been observed to extend over 50 km
downwind of a wind farm, particularly during stable conditions when little atmospheric turbulence is present to erode the wake
(Christiansen and Hasager, 2005; Platis et al., 2018).

Wind farm wakes have been observed to impact their ambient environment, particularly at night via turbine-induced mixing
of the nocturnal inversion. Baidya Roy and Traiteur (2010) first used in-situ observations within a wind farm to identify that a
net downward transport of warm air at night causes a statistically significant increase in surface temperature in stably stratified
boundary layers. Further in-situ measurements of lidar profiles and surface fluxes from the Crop Wind Energy Experiment
(CWEX) over an Iowa utility-scale wind farm indicate that turbines modify the flow fields and surface heat flux above and
below the rotor layer, causing average transient surface temperature increases between 0 to 0.5 K overnight, reaching 1.5 K
in some cases (Rajewski et al., 2013, 2014, 2016). Another field campaign by Smith et al. (2013) in a large wind farm in
the United States Midwest identify a strong surface warming (1 to 1.5 K) in the wake of the wind farm at night, with no
substantial warming or cooling signals during the daytime. Armstrong et al. (2016) find that operational wind turbines raised
nighttime air temperature by 0.18 K and absolute humidity by 0.03 g m$^{-3}$ at a peatland wind farm in Scotland. Satellite-derived
analyses also report surface warming (up to 1 K) within wind farms and also up to 5 km downwind (Zhou et al., 2012; Xia
et al., 2016). In-situ airborne measurements of offshore wind farms also find temperature increases of 0.5 K and water vapor
decreases of 0.5 g kg$^{-1}$ in the rotor layer downwind during stably stratified conditions (Platis et al., 2018; Siedersleben et al.,
2018a). Measurements from two 120-m tall towers in Iowa detect differences in the timing of nocturnal transitions due to the
presence of a wind farm, where a single turbine wake decoupled turbulent connection between the surface and above the wind
turbine, accelerating the onset of near-surface stabilization by a few hours and lengthening the transition period by up to an
hour (Rajewski et al., 2020).

Due to sparse data from operational wind farms, modeling studies are often used to examine local and regional impacts of
existing and hypothetical wind farms. Baidya Roy (2004) conducted the first study within a coupled land-atmosphere mesoscale
model, representing wind turbines as sinks of momentum and sources of turbulence to demonstrate that the simulated wind
farm slows hub-height winds and generates turbulent eddies that enhance vertical mixing, leading to a warming and drying of
the surface air. Wind turbines can also be represented numerically in mesoscale simulations by exaggerating surface roughness
to represent the local reduction of wind speed of wind farm wakes (Keith et al., 2004; Frandsen et al., 2009; Barrie and Kirk-
Davidoff, 2010; Fitch, 2015). This enhanced surface roughness approach was later shown to produce erroneous predictions,
including the wrong sign of surface temperature change through the diurnal cycle (Fitch et al., 2013). Recent mesoscale
modeling studies have used the turbine power and thrust curves to define the elevated momentum sink and turbulence generation
of a simulated wind turbine (Fitch et al., 2012). The turbine power and thrust curves give the manufacturer-specified relationship
between hub-height inflow wind speed, power generation, and force exerted onto the ambient air by a specific wind turbine.
These specifications can better predict meteorological impacts of wind turbines from hub height to the surface and form the



basis for multiple wind farm parameterizations in mesoscale numerical weather prediction models, including the Wind Farm Parameterization (WFP) (Fitch et al., 2012; Fitch, 2016).

The open-source WFP of the Weather Research and Forecasting (WRF) model collectively represents wind turbines in each model grid cell as a momentum sink and a turbulence source within the vertical levels intersecting the turbine rotor disk (Fitch et al., 2012; Fitch, 2016). The virtual wind turbines convert kinetic energy from the wind into power, which is reported as an aggregate sum in each model grid cell. The default setting of the WFP dictates that the turbine-induced turbulence generation is derived from the difference between the power and thrust coefficients, and this option must remain enabled in order to produce

the vertical mixing necessary to attain the expected nocturnal surface warming (Tomaszewski and Lundquist, 2020). Users can adjust the specifications of the parameterized turbine, including its rotor diameter, hub height, thrust coefficients, and power curve, as well as its latitude and longitude location. WFP simulations have been validated with power production data (Lee and Lundquist, 2017a) and airborne measurements of winds (Siedersleben et al., 2018b), temperature and moisture (Siedersleben et al., 2018a), and turbulence (Siedersleben et al., 2020) and have reproduced the observed localized, nighttime, near-surface

warming produced by wind turbines mixing warmer air from the nocturnal inversion down to the surface (Fitch et al., 2013; Cervarich et al., 2013; Lee and Lundquist, 2017b; Xia et al., 2017, 2019). To our knowledge, the WRF WFP has not yet been applied to explore interactions between wind farms and transient phenomena like thunderstorm outflow boundaries.

A thunderstorm gust front, or outflow boundary, marks the advancing surface boundary of the outflow of an evaporatively cooled downdraft from a thunderstorm (e.g., Goff, 1976; Droegemeier and Wilhelmson, 1987). Outflow boundary passage is

often associated with a significant change in surface meteorological conditions, including a sharp decrease in temperature, a pronounced wind direction shift, and damaging straight-line winds fueled by a strong horizontal pressure gradient across the outflow boundary line (Wakimoto, 1982). The propagation of outflow into a thermodynamically favorable ambient environment can initiate convection far from the source thunderstorm (Carbone et al., 1990), and the strong low-level wind shear associated with propagating outflows has caused several aircraft accidents (Zrnic and Lee, 1983) that would otherwise be unlikely in a

typical wind farm wake environment (Tomaszewski et al., 2018). Outflow boundary propagation is thus of interest in short-range severe weather and aviation forecasting and nowcasting.

Doppler radar observations can provide kinematic information of the full depth of thunderstorm outflow (e.g., Wakimoto, 1982; Klingle et al., 1987; Mueller and Carbone, 1987; Quan et al., 2014), with an outflow boundary's presence and propagation speed often identified via a "fine line" in radar reflectivity. On June 18, 2019 around 0100 UTC, National Weather

Service radar reflectivity indicated the presence of thunderstorm-generated outflow propagating east-southeast north of Lubbock, Texas. A section of the outflow boundary that encountered a wind farm experienced a notable reduction in propagating speed, qualitatively suggesting that the wind farm impacted the outflow boundary progression.

Here we use a numerical weather prediction mesoscale model capable of simulating outflow boundary propagation to explore its interaction with a parameterized wind farm during the aforementioned June 2019 event. Numerical models have previously

been utilized to gain insight into the life cycle and dynamics of thunderstorm outflow (e.g., Droegemeier and Wilhelmson, 1987), and more recently in the Weather Research and Forecasting model by Duda and Gallus (2013) and Nugraha and Trilaksono (2018) with success. The inclusion of a Wind Farm Parameterizaton (WFP) in WRF (e.g., Fitch et al., 2012; Fitch, 2016)





to capture wind farm-near environment interactions makes this model a favorable tool for such a study considering the impacts wind farms may have on outflow boundaries and their resulting changes in temperature, wind, and precipitation.

We hypothesize that wind farms can modify transient and mesoscale features like thunderstorm outflow boundaries. Section 2 describes the case study and the model setup. Section 3 presents the modifications to the outflow propagation by the wind farm and the impacts to surface temperature, winds, moisture, and precipitation. Section 4 summarizes our results confirming the WRF WFP and radar data capture the wind farm modifying the outflow.

## 2  Methodology

### 2.1  Case Description


The 18-19 June 2019 outflow event near Lubbock, Texas is highlighted in this study as the first known and archived case of an outflow boundary passing over and being modified by a wind farm, brought to our attention on social media by Jessie McDonald (@jmeso212). The event began with a cluster of thunderstorms propagating eastward over eastern New Mexico and the western Texas panhandle. These storms formed an organized mesoscale convective system (MCS) around 2300 UTC on

June 18 at the New Mexico/Texas border and shifted to move southeastward. An outflow boundary originated from this MCS, visible as a fine line on NEXRAD WSR-88D displays beginning at approximately 2340 UTC (Fig. 1a). This outflow boundary advanced southeastward out ahead of the MCS, eventually reaching the Hale wind farm at 0050 UTC on June 19. The wind farm can be detected on the radar display (Fig. 1a) as a cluster of speckled points of high reflectivity, indicative of the hard-target echos of radar beams reflecting off of spinning turbines, known as wind turbine clutter (Isom et al., 2009). A defined notch

appeared within the outflow boundary immediately following passage over the wind farm, suggesting a significant reduction in propagating speed where the outflow encountered and interacted with the wind farm (Fig. 1b,c).

### 2.2  Observations Available

The National Weather Service NEXRAD WSR-88D radar in Lubbock, Texas (KLBB) (Klazura and Imy, 1993) provides the initial visual of the outflow propagating and interacting with the wind farm during the June 2019 event. Level II radar data

(e.g., base reflectivity, base velocity) are provided by the NOAA National Centers for Environmental Information (NOAA National Weather Service, 1991) at 4-minute temporal resolution and quantify the speed and position of the outflow boundary throughout the event.

Surface observations are available through the West Texas Mesonet, a statewide observation network consisting of 40 automated surface meteorological stations that measure up to 15 meteorological parameters over an observation period of 5 minutes

(Schroeder et al., 2005). A 5-minute observation reporting time has been previously proven sufficient in resolving other density current passages (Toms et al., 2017). The Abernathy surface station is located 5 km southwest of the wind farm in our study (grey diamond in Fig. 3) and provides 5-minute resolution validation data of 1.5-m temperature, 10-m wind speed and direction, and 1.5-m humidity, among other variables, for the precursor outflow state prior to wind farm interaction for our





simulations. We explored accessing meteorological information from the Hale wind farm and others in the vicinity, but those
data are proprietary and not available.

## 2.3 Simulations Conducted

We conduct the simulation comprising our study with version 3.8.1 of the Advanced Research WRF (ARW) model (Skamarock
and Klemp, 2008; Powers et al., 2017). We define a simulation with three nested domains with horizontal resolutions of 27,
9, and 3 km, respectively, where the innermost, 3-km domain is centered over the wind farm and outflow event location (Fig.
2a). Our previous investigation (Tomaszewski and Lundquist, 2020) of the sensitivity of the WRF WFP to spatial resolution
suggests that 3-km horizontal resolution is adequate for resolving the wind farm effects. Also based on the results of that study,
which argue that the WFP requires fine vertical resolution near the surface, we set the vertical resolution to be ∼10 m in the
lowest 200 m (Fig. 2b), stretching vertically thereafter, for a total of 58 vertical levels between the surface and 170 hPa. The
model time step is 30 s on the outer domain, refined by a factor of 3 for each nest. Turbine-induced turbulence is parameterized
via a source of TKE. The $0.7°$ ERA-Interim (ECMWF, 2009; Dee et al., 2011) data set provides initial and boundary conditions
for the simulations, and topographic data are provided at 30-s resolution (nominally 0.8 km at this latitude). Physics options
include the Dudhia short-wave radiation (Dudhia, 1989) with a 30-s time step, the Rapid Radiative Transfer Model (RRTM)
long-wave radiation scheme (Mlawer et al., 1997), a surface layer scheme that accommodates strong changes in atmospheric
stability (Jimenez et al., 2012), the second order Mellor-Yamada-Nakanishi-Niino (MYNN2) PBL scheme (Nakanishi and
Niino, 2006) *without* TKE advection, land surface physics with the Noah Land Surface Model (Ek et al., 2003), the single-
moment 6-class microphysics scheme (Hong and Lim, 2006), and the explicit Kain–Fritsch cumulus parameterization (Kain,
2004) on domains with horizontal resolutions coarser than 3 km. We simulate the 6-hour window around the time when the
outflow passed over the wind farm (June 18 2200 UTC to June 19 0400 UTC). We begin spinup 10 hours prior, at 1200 UTC
on June 18.

The USGS Turbine Database (Hoen et al., 2020) provides the latitude-longitude model input locations of the wind turbines at
the Hale wind farm (Fig. 2c,d). We use power and thrust curves from the 1.5-MW Pennsylvania State University (PSU) generic
turbine (Schmitz, 2012), based on the General Electric SLE turbine (80-m hub height and 77-m rotor diameter). This turbine
model closely matches the 2-MW Vestas turbines actually installed at the wind farm of interest, and Siedersleben et al. (2018b)
show little sensitivity to the exact turbine power curve. We assess the impact the wind farm has on the model solution of the
outflow by comparing a simulation without the WFP to a simulation with the WFP, as in Fitch et al. (2012); Lee and Lundquist
(2017a); Lundquist et al. (2018), and Redfern et al. (2019). We specifically investigate differences in the near-surface wind
speed, temperature, moisture, and precipitation solutions between simulations with and without the wind farm as the storm and
associated outflow pass over the wind farm domain.

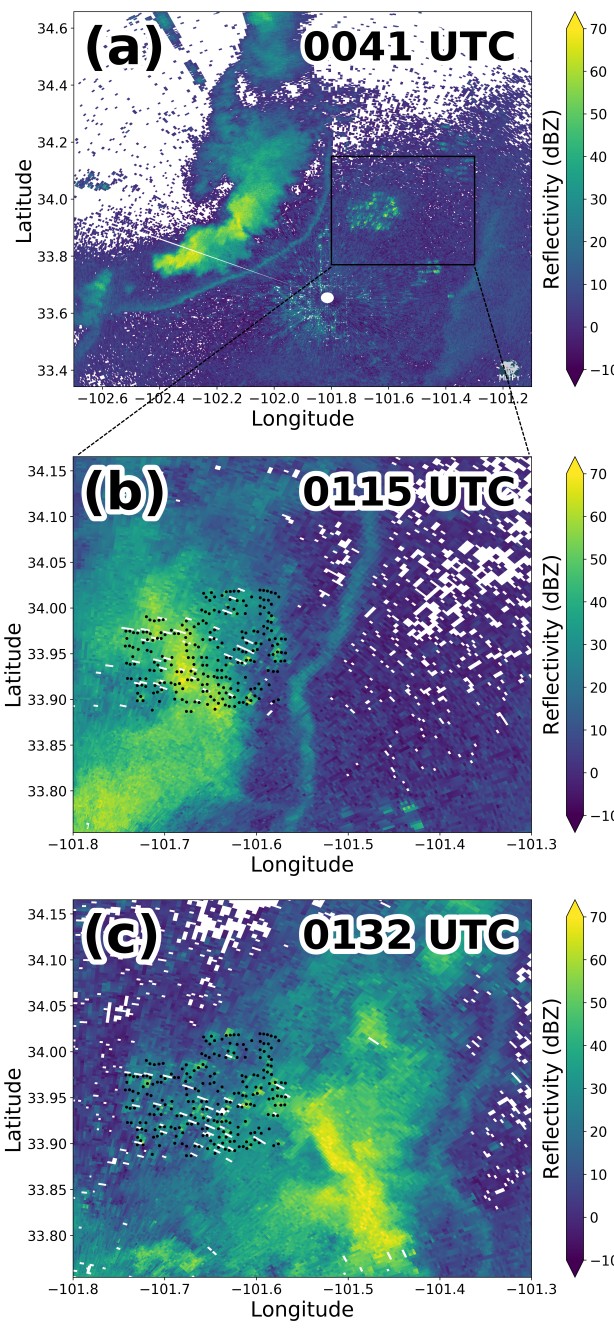

**Figure 1.** NEXRAD WSR-88D radar reflectivity from the Lubbock radar site (KLBB) (a) prior to the outflow boundary arriving at the wind farm, (b) immediately following outflow boundary passage over the wind farm, and (c) several minutes after passage. Panels (b) and (c) are zoomed in closer to better view the shape of the boundary, and that subset is denoted in panel (a) by the black box.

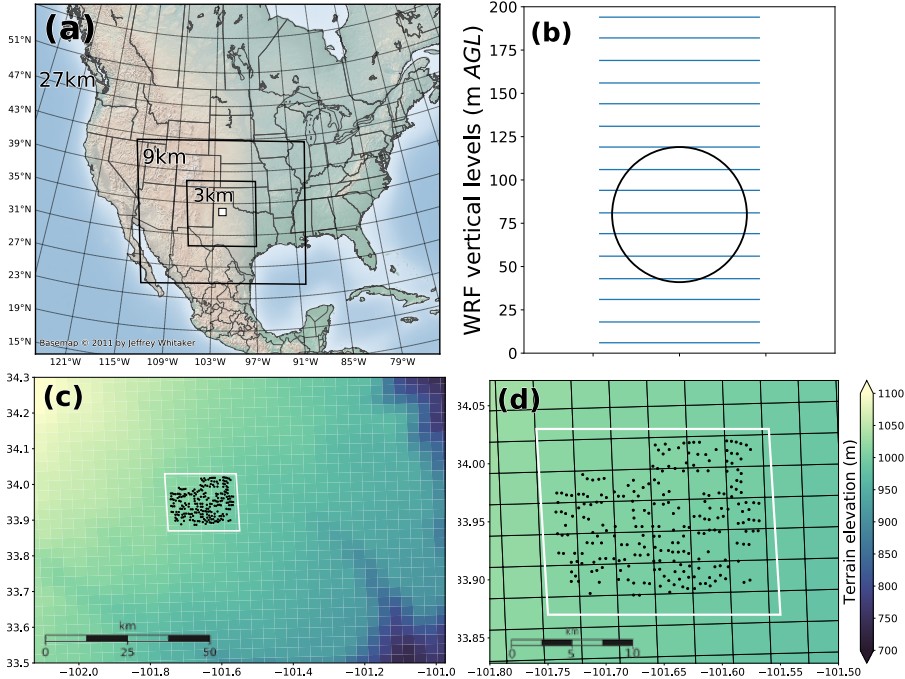

**Figure 2.** Maps representing (a) the nested domains for the simulations, with the Hale wind farm denoted by the white square, (b) the model vertical levels intersecting the simulated turbine rotor (black circle), (c) the topography around the Hale wind farm, and (d) the wind turbine layout over the same terrain contour as in (c). The white polygon in panels (c) and (d) denotes the approximate outline of the Hale wind farm. Geography data are provided by Matplotlib's (Hunter, 2007) Basemap © 2011 by Jeffrey Whitaker.

## 3   Results

### 3.1   Performance of WRF against observations


The WRF simulation with the Wind Farm Parameterization (WFP) produces reasonable solutions of the thunderstorm outflow event on June 18-19, 2019. Three consecutive top-down views of 2-m temperature within a section of the domain illustrate the outflow progression (Fig. 3). By 22:36 UTC on June 18, a cold pool had developed from the thunderstorm downdraft, forming the outflow (Fig. 3a). This outflow propagated southeast toward the wind farm, eventually passing over it by 23:22 UTC (Fig.

3b), and spreading further southeast by the end of the UTC day (Fig. 3c). The spatial coverage and shape of the outflow reasonably match the shape of the outflow boundary visible on radar (Fig. 1). However, the outflow event in the simulation occurs too early. The simulated outflow passes over the wind farm location by 23:22 UTC on June 18, whereas the radar indicates this passage occurs around 0115 UTC on June 19, about 2 hours later (Fig. 1b). This discrepancy could be due to the boundary conditions or other model configuration choices, though as previously mentioned, the structures of the simulated and

observed cold pools are similar and therefore the simulation suits the needs of the study.



To better understand WRF's skill in simulating the intensity of the outflow event, we plot a time series from the nearby Abernathy West Texas Mesonet surface station against that from the corresponding point in the model domain (grey triangle in Fig. 3). Model results are shifted ahead 2 hours to allow for direct comparison between the simulation and observations during the outflow passage, as done in the Arthur et al. (2020) investigation of a frontal passage. The WRF simulation (solid lines in

Fig. 4) predicts similar 10-m wind speeds as observed (dotted lines) before the passage in addition to an accurate magnitude of wind speed increase associated with the outflow arrival. The simulated winds remain elevated near ∼23 m s$^{-1}$ for 15 minutes before decreasing close to the prefrontal state, while the surface station observations decrease almost immediately, possibly an artifact of the 5-min sampling in the observations as opposed to the 1-min sampling in the simulation (Fig. 4a). The simulation displays biases in the 2-m temperature and moisture precursor states (Fig. 4b,c). WRF initially has a 2.5 K warm bias, a

∼35% relative humidity (RH) dry bias, and ∼3 g kg$^{-1}$ dry bias against the observations. These model biases could be due to inaccuracies in the soil moisture that stem from differences in precipitation that occurred earlier in the day. The magnitude of the 2-m temperature decrease (Fig. 4b) and moisture increases (Fig. 4c) due to the outflow arrival in WRF seem adequate, albeit slightly more intense than in the observations.

### 3.2 Differences in outflow passage between wind farm and no wind farm simulations

Having validated WRF's ability to adequately capture the outflow event, we next compare the two WRF simulations to assess the impact a parameterized wind farm has on the simulated outflow. Three instantaneous map views show the difference in 2-m temperature between the simulations, with the no wind farm (NWF) case subtracted from the wind farm parameterization (WFP) case (Fig. 5). Regions of cooler temperatures (blue) indicate that the temperature in the WFP simulation is cooler than in the NWF simulation, suggesting faster movement of the outflow bringing cooler temperatures. Conversely, red regions indicate

warmer temperatures in the wind farm-containing simulation, indicating the outflow is moving slower in this simulation than in the NWF simulation. Early in the outflow event, only subtle differences exist between the simulations upwind from the wind farm (Fig. 5a). These differences increase in magnitude after the outflow passes over the wind farm. A compact region of warmer temperatures (up to 8 K) in the wind farm simulation emerges following outflow passage over the wind farm, indicating that interaction with the wind farm has caused that section of the advancing outflow to slow its propagation speed (Fig. 5b).

This region of slowed outflow expands in area as the outflow progresses southeastward (Fig. 5c). A similar speed reduction is visible in the bent outflow shape of the radar observations (Fig. 1b,c). The wind farm also appears to act as a barrier around which the outflow channels and accelerates, most notably immediately following passage over the wind farm (Fig. 5b).

We next sample a point from both the WFP and NWF simulations downwind of the wind farm location (white diamond in Fig. 5) to assess how differences between the simulations evolve at that point following outflow passage (Fig. 6). Close

agreement exists between the simulations across all variables plotted preceding arrival of the outflow. Upon outflow approach, the 10-m wind speed increases first in the NWF simulation (dashed line), reaching a maximum of ∼30 m s$^{-1}$. The WFP simulation (solid line) begins its outflow-induced increase a few minutes after the NWF simulation and attains a smaller initial wind speed maximum of ∼25 m s$^{-1}$. A secondary pulse of increased wind speeds occurs in both simulation cases and reaches

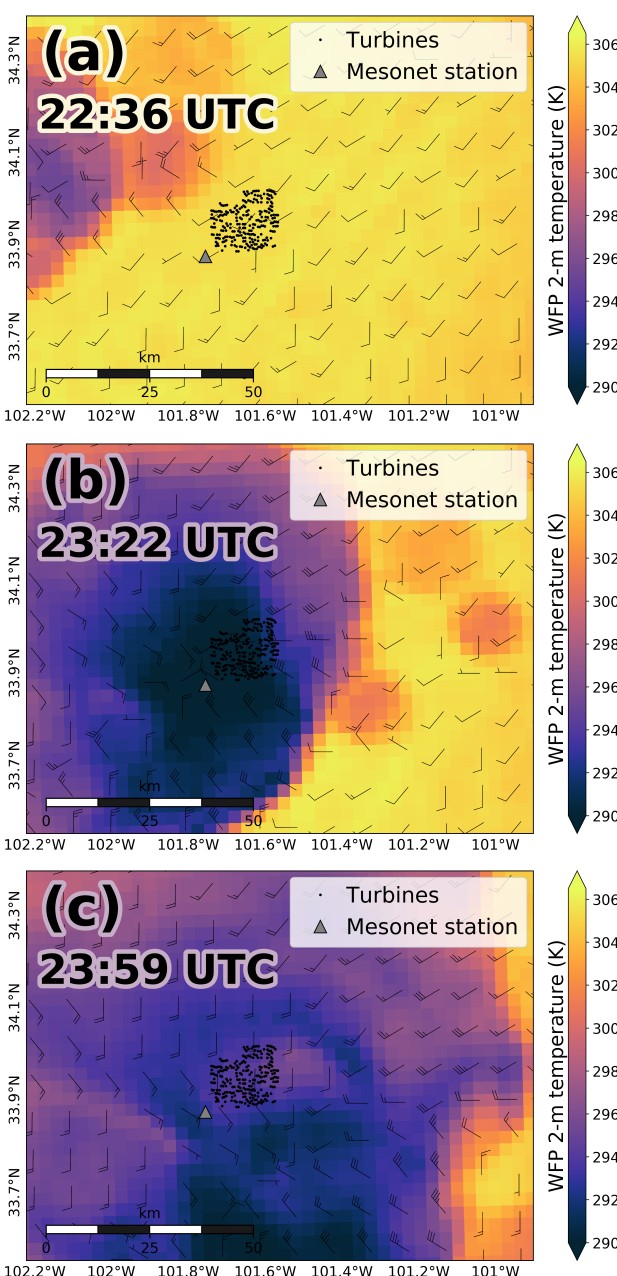

**Figure 3.** Snapshots of simulated 2-m temperatures from the Wind Farm Parameterization (WFP) simulation (a) prior to the outflow boundary arriving at the wind farm, (b) immediately following outflow boundary passage over the wind farm, and (c) several minutes after passage. Wind barbs are shown in kts.

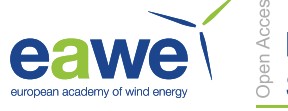

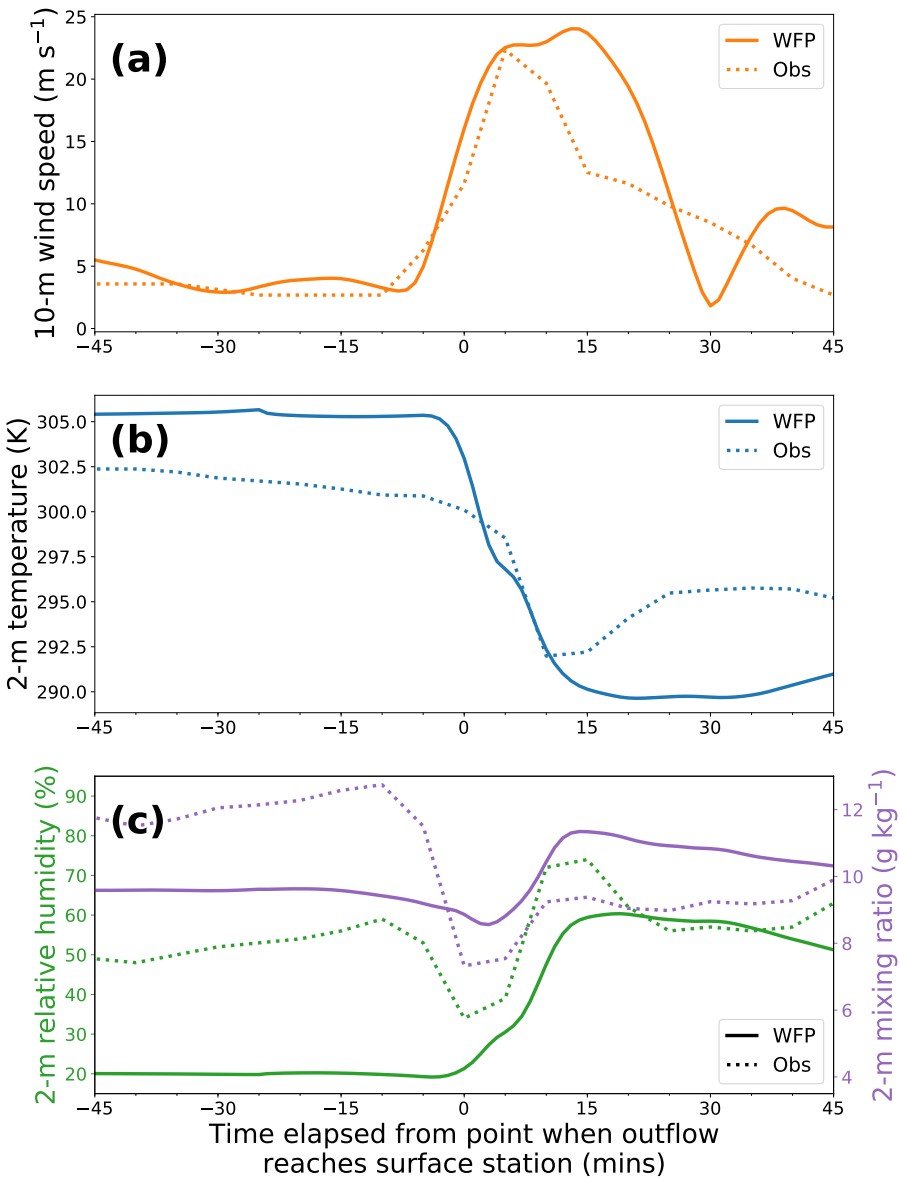

**Figure 4.** Time series comparing the Wind Farm Parameterization (WFP) simulation solutions against Mesonet surface station observations of (a) 10-m wind speed, (b) 2-m temperature, and (c) 2-m relative humidity and mixing ratio.



similar magnitudes, suggesting the modified outflow in the wind farm case recovers after the initial disruption by the wind farm
(Fig. 6a).

The temporal evolution of 2-m temperature is similar to that of the wind speed. The WFP and NWF simulations produce the same initial temperature until the WFP simulation diverges from the NWF simulation due to the wind farm-modified outflow approaching ∼3 minutes later. The associated outflow cooling is of similar magnitudes (∼12.5 K) between the simulations, but the WFP simulation reaches its minimum temperature ∼4 minutes after the NWF simulation (Fig. 6b).

Differences in the 2-m relative humidity between the simulations evolve similarly as those in the 2-m temperature. Both simulations maintain a value near ∼20% until the passing outflow causes an increase up to 50%, with the increase occurring for the WFP simulation 4-5 minutes after the NWF (green lines in Fig 6c). The absolute moisture quantity (2-m mixing ratio, purple lines) experiences a greater lag in outflow-induced increase compared to the other variables. The WFP simulation reaches its peak in moisture (10.5 g kg$^{-1}$) ∼6 minutes after the NWF does. Compared to the 2- or 3-minute delay in WFP
outflow response in the temperature and wind speeds, the longer delay for the mixing ratio response indicates that the wind farm-modified outflow impacts absolute moisture more, suggesting that the change in moisture due to the outflow lags the change in temperature.

We corroborate the proxies for outflow propagation speed in the time series of meteorological variables (Fig. 5) by directly quantifying the speed of the simulated and observed outflow boundaries (Fig. 7a). We measure the observed outflow propaga-
tion speed by tracking the reflectivity fine-line along a transect and recording its distance traveled every data update (typically 4 minutes). Without a fine-line present in the simulations to denote the outflow boundary, we choose to track the simulated outflow using the spatial gradient in wind speed, specifically the $\frac{4\,\mathrm{m\,s^{-1}}}{\mathrm{km}}$ contour (e.g., Fig. 7b). The simulations are examined at 4-minute intervals to match the temporal resolution of the radar data. Both simulation and radar outflow are measured against a 5x5 km grid to estimate distance traveled (see Fig. 7b). The transect along which we measure distance traveled is oriented
to track through the wind farm and the region of maximum outflow distortion by the wind farm. Three separate measurement examinations are conducted for each case (i.e., radar, WFP simulation, and NWF simulation) to account for human error. The averages of each case are plotted in Fig. 7a, around which ±1 standard deviation forms the shaded cloud and serves as our error bounds. As in Fig. 4 and Arthur et al. (2020), the simulation results are shifted forward 2 hours to align with the radar results. A running average with an 8-minute window was applied to all three time series to smooth the results for viewing.

As suggested in Fig. 5 and Fig. 6, propagation speeds of both simulation cases and the radar data begin at similar speeds near 80 km hr$^{-1}$ (Fig. 7a). The simulated and observed outflows decelerate slightly as they propagate away from the source thunderstorm. When the radar outflow (blue line) encounters the wind farm, its speed reduces from 60 km hr$^{-1}$ to nearly 40 km hr$^{-1}$. The radar outflow recovers within 10 minutes back to >60 km hr$^{-1}$ before being obscured by precipitation. Similarly, the Wind Farm Parameterization (WFP) simulation (orange line) fluctuates around 70 km hr$^{-1}$ until encountering the wind
farm, when it then drops in speed to about 40 km hr$^{-1}$. The WFP simulation experiences a larger reduction in speed than observed but reaches its speed minimum ∼8 minutes later than the observations. Additionally, the WFP simulation recovers its speed twice as slowly as the observed outflow. Such delays in the WFP outflow evolution could be artifacts of the 3-km model




resolution. The no wind farm (NWF) simulation (green line), lacking wind farm interference, maintains a propagating speed between 60 and 75 km hr$^{-1}$ throughout the period of interest.

## 3.3 Simulated impacts of modified outflow boundary to precipitation

Subtle but significant impacts of wind farm-modified outflow to meteorological variables like wind speed, temperature, and moisture outlined in Sec. 3.2 prompt the question of the extent to which a wind farm-modified outflow boundary can impact precipitation location and quantity. We address this question by integrating the total precipitation over a 100-km radius around the wind farm and comparing these quantities for the WFP (green line) and NWF (black dashed line) simulations every minute (Fig. 8a) and accumulated in time (Fig. 8b) over 3 hours. While the 1-minute precipitation totals across the region differ slightly between the simulations, the total accumulated precipitation remains unchanged despite the altered outflow in the WFP case. We conclude that the introduction of roughness elements may change the distribution of the precipitation by a maximum of ∼1 cm across the domain at a single moment in time (Fig. 8a), but the overall precipitation accumulation is unaffected (Fig. 8b). Furthermore, a histogram detailing the number of 3-km grid cells that do experience a change in precipitation at a 1-minute moment in time over 3 hours due to the presence of the wind farm reveals that no single grid cell experiences a delta greater than ±7 mm, and over 93.4% of grid cells experience 0 change in precipitation (Fig 9). Changes to precipitation due to the wind farm are thus both transient and localized.

## 3.4 Power production at the simulated wind farm

Given that wind farms can modify outflow and their associated meteorology, we next explore the effects an incoming outflow can have on a wind farm and its power production. Time series of the simulated 80-m wind speeds from all turbine-containing grid cells in the Wind Farm Parameterization (WFP) simulation indicate that several grid cells exceed the wind turbine's cut-out speed (25 m s$^{-1}$, black dashed line in Fig. 10a), most notably at 23:10 and 23:35 UTC. Winds in excess of this cut-out speed force the turbines to brake their blades to prevent structural damage, halting power generation. The corresponding time series of power from turbine-containing grid cells and the total integrated farm power (Fig. 10b) reflect this reduction in power during those times when the cut-out wind speed is reached. Power data from the Hale wind farm are proprietary and unavailable for validation, though simulation data suggest outflow winds are high enough to cause wind turbines to cut out and reduce total farm power generation (Fig. 10).

## 4 Discussion and Conclusions

Increasing deployment of wind energy necessitates further knowledge on the environmental impacts of wind farms to ensure their long-term sustainability and suitability. A lower-atmospheric phenomenon not yet explored in relation to interacting with wind energy is thunderstorm outflow. Herein, we assess the impact a wind farm can have on outflow propagation via observations and simulations.

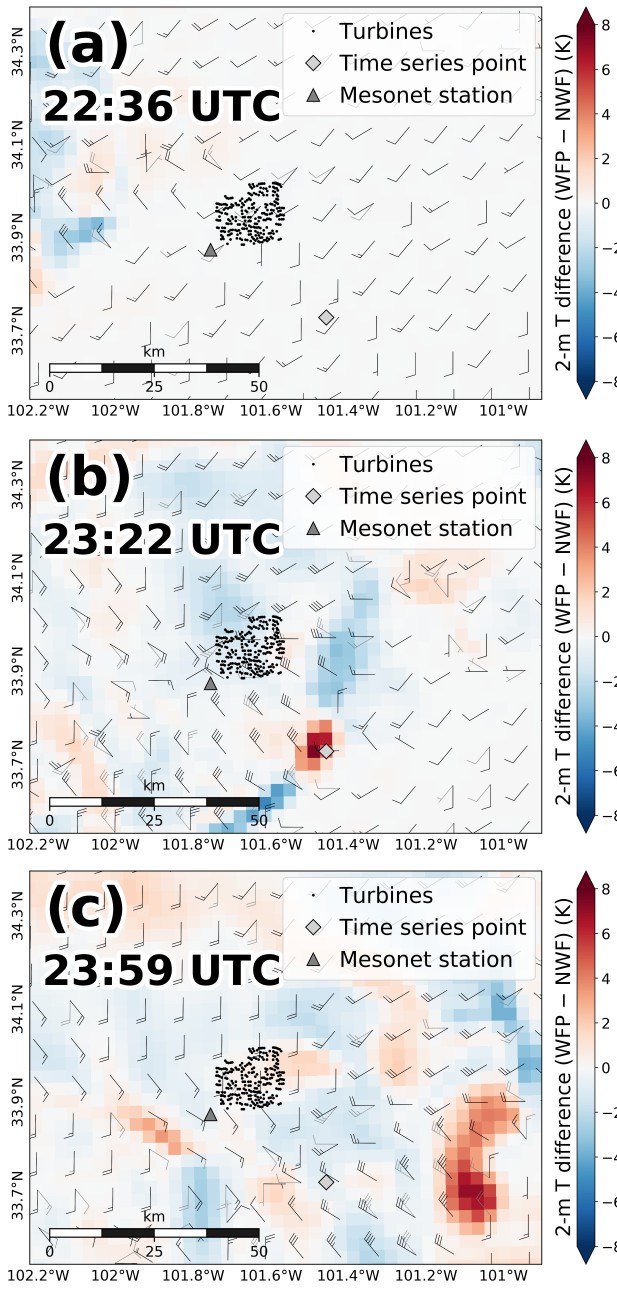

**Figure 5.** Snapshots of differences in 2-m temperature between the wind farm parameterization (WFP) and no wind farm (NWF) simulations (a) prior to the outflow boundary arriving at the wind farm, (b) immediately following outflow boundary passage over the wind farm, and (c) several minutes after passage. Wind barbs are shown in kts, with the darker (lighter) barbs representing the WFP (NWF) winds.

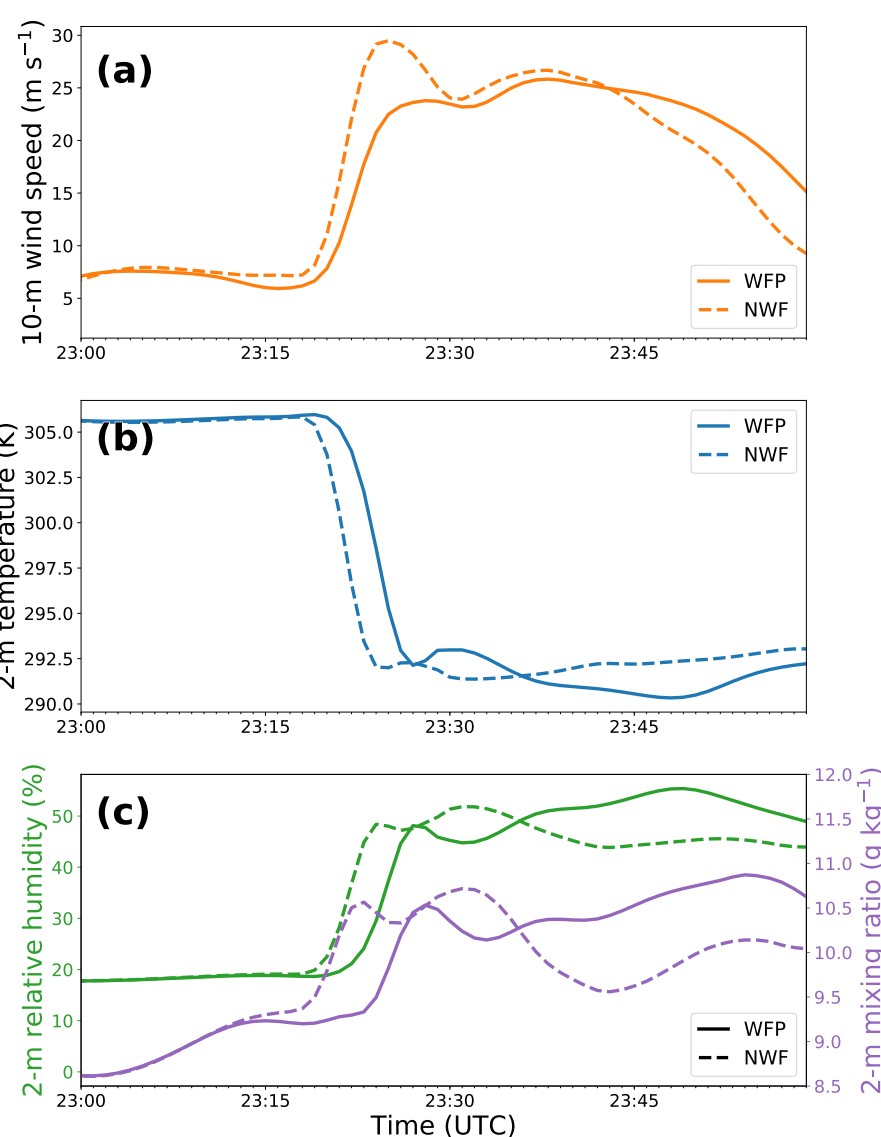

**Figure 6.** Time series comparing the Wind Farm Parameterization (WFP) and no wind farm (NWF) simulation solutions of (a) 10-m wind speed, (b) 2-m temperature, and (c) 2-m relative humidity and mixing ratio.

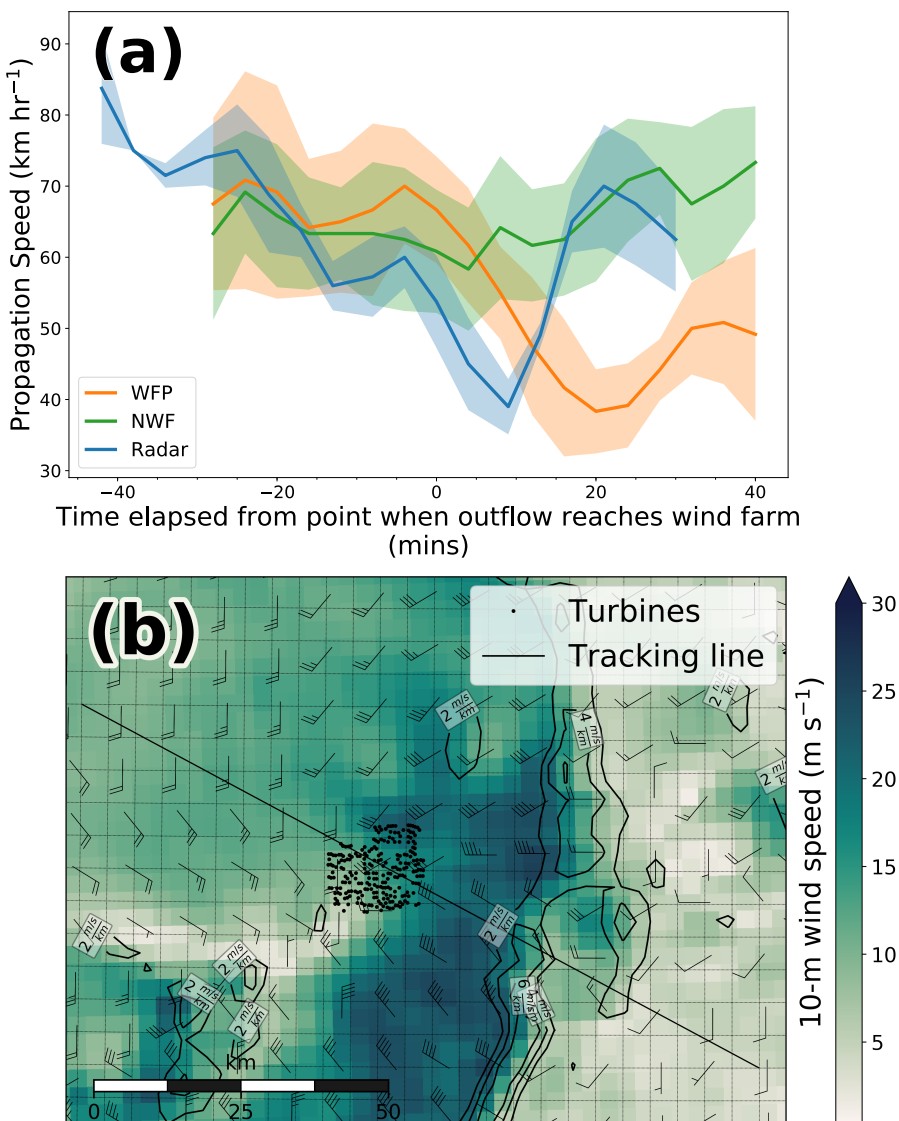

**Figure 7.** Plots (a) comparing the propagation speed of the outflow in time between the Wind Farm Parameterization simulation, no wind farm (NWF) simulation, and radar observations. The schematic in (b) shows the process for calculating the simulation outflow speed, where the line through the domain shows the transect along which speed was measured, with the gradients in 10-m wind speed providing the position of the boundary to track.

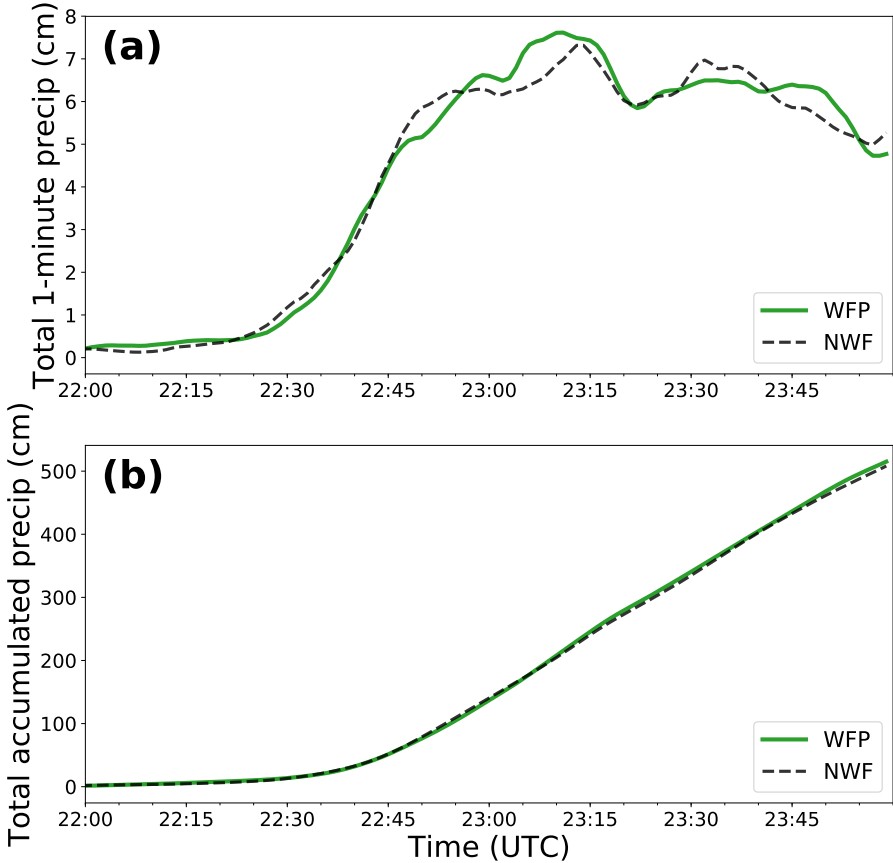

**Figure 8.** Time series comparing the Wind Farm Parameterization (WFP) and no wind farm (NWF) simulation solutions of (a) total 1-minute precipitation and (b) total accumulated precipitation, both quantities integrated spatially across a region of 100-km radius around the Hale wind farm.

We first observed wind farm impacts to outflow in NEXRAD WSR-88D radar reflectivity. On June 18, 2019, a section of an advancing outflow boundary visible on radar encountered the Hale wind farm near Lubbock, Texas and decelerated in response.

We ran two Weather Research and Forecasting (WRF) simulations to capture this event: one with a Wind Farm Parameterization (WFP) enabled and another with no wind farm present (NWF). Using observations from a West Texas Mesonet surface station, we verified that the simulations were producing reasonable solutions of the outflow event and could be used to quantify the extent a wind farm can modify propagating outflow.

Just as with the radar reflectivity, spatial differences between the WFP and NWF simulations exhibited a similar pattern

indicating the wind farm slowed the progress of the outflow boundary (Fig. 1, Fig. 5). Time series of simulated surface wind speed, temperature, and moisture revealed that perturbations associated with outflow passage were delayed by up to 6 minutes when the wind farm was present in the simulation (Fig 6.) Approximations of outflow propagation speed in the radar and





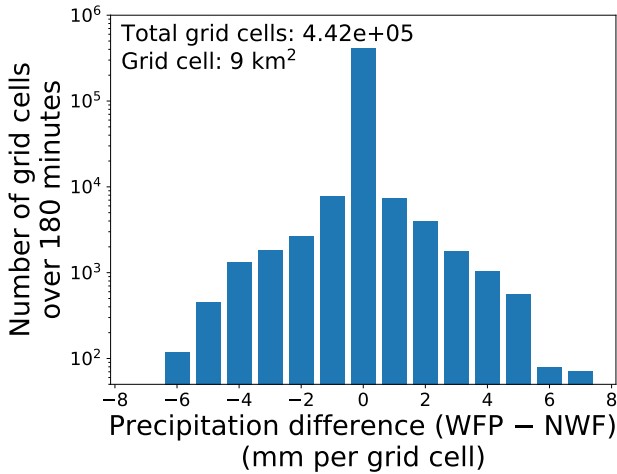

**Figure 9.** Histogram detailing the 1-minute differences between the Wind Farm Parameterization (WFP) and no wind farm (NWF) simulations in precipitation at each grid cell within a 100-km radius of the Hale wind farm over 180 minutes.

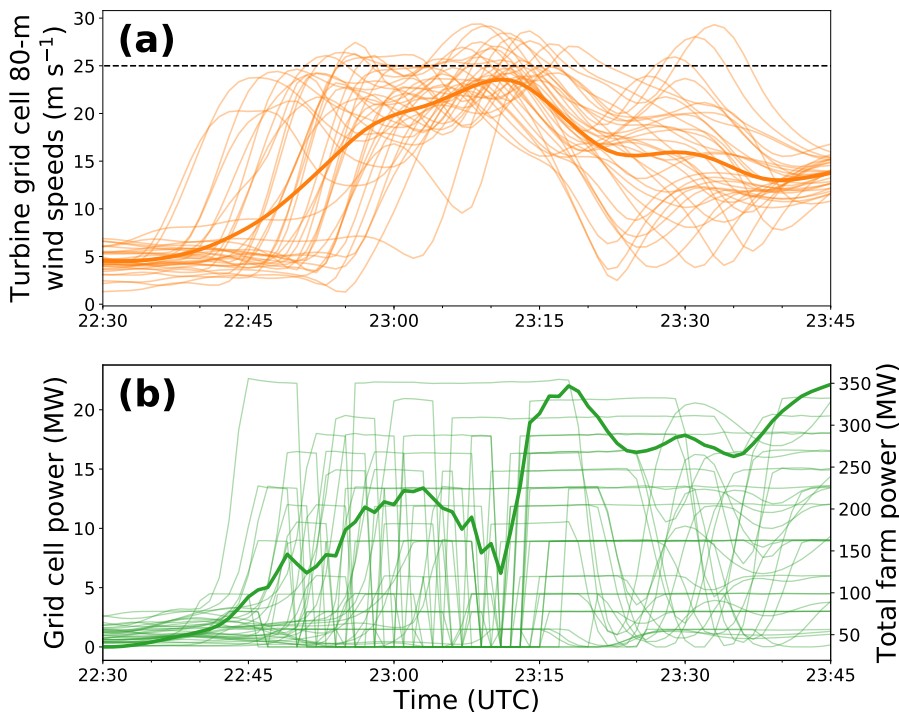

**Figure 10.** Time series of simulated (a) 80-m wind speeds from all turbine-containing grid cells (average wind speed plotted in the thicker line) and (b) power from all turbine-containing grid cells (total farm power production plotted in the thicker line) in the Wind Farm Parameterization (WFP) simulation.





simulation data confirmed that both the radar-observed and WFP simulation outflows experienced speed reductions of over 20 km hr$^{-1}$ after encountering the wind farm, while the NWF simulation maintained near constant speed throughout the period

(Fig. 7). Impacts to precipitation were minimal, with no change to total accumulation across the domain (Fig 8). Localized shifts to precipitation location in the WFP simulation caused a maximum instantaneous grid-cell precipitation difference of 7 mm km$^{-2}$, but 93.4% of grid cells within the area over the event period experienced no change in precipitation (Fig. 9).

While we have shown that a wind farm can interact with and modify thunderstorm outflow, impacts to the modified outflow propagation speed and associated kinematic and thermodynamic variables are transient and localized. These subtle changes

arising from wind farm interaction may be useful to consider when conducting nowcasting of precipitation and wind speed on a scale of a few kilometers and minutes, perhaps for aviation or other time-sensitive purposes. Impacts beyond that scale appear to be negligible.

This study uses a single known case of a wind farm interacting with outflow and is corroborated by simulations of that case. This case study could motivate a larger-scale climatology of additional outflow-wind farm interaction events. Such a

climatology could consider wind farms of different layout and sizes, as well as different turbine types and sizes to assess generalized sensitivity of atmospheric modifications to the turbine layouts and density. In previous studies (Lundquist et al., 2018), we have noticed wind farms apparently modifying the passage of frontal boundaries, so a large-scale climatology of such events, tracking frontal propagation speed, could shed more light on how widespread and impactful the modification of atmospheric processes by wind farms can be.

*Code and data availability.*  The WRF-ARW model code (https://doi.org/doi:10.5065/D6MK6B4K) is publicly available at http://www2.mmm.ucar.edu/wrf/users/. This work uses the WRF-ARW model and the WRF Preprocessing System (WPS) version 3.8.1 (released on 12 August, 2016), and the wind farm parameterization is distributed therein. Initial and boundary conditions are provided by Era-Interim (Dee et al., 2011) available at https://rda.ucar.edu/datasets/ds627.0/. Topographic data are provided at a 30-s resolution from http://www2.mmm.ucar.edu/wrf/users/download/get_source.html. The PSU generic 1.5-MW turbine (Schmitz, 2012) is available at

https://doi.org/10.13140/RG.2.2.22492.18567. The user input and data needed to recreate the figures and analysis are located at https://doi.org/10.5281/zenodo.3765421.

*Author contributions.*  JKL and JMT conceived the research and designed the WRF simulations; JMT carried out the WRF simulations and wrote the manuscript with significant input from JKL.

*Competing interests.*  The authors declare that they have no competing interests.



*Acknowledgements.*    This work and JMT were supported by an NSF Graduate Research Fellowship under grant number 1144083. WRF simulations were conducted using the Extreme Science and Engineering Discovery Environment (XSEDE), which is supported by National Science Foundation grant number ACI1053575. JKL's effort was supported by an agreement with NREL under APUP UGA-0-41026-65. We thank Jessie McDonald (@jmeso212) for identifying this event and catalyzing the interesting discourse on Twitter that inspired this research.



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
