# Peer review of "Observations and Simulations of a Wind Farm Modifying a Thunderstorm Outflow Boundary"

_Wind Energy Science, 2020_

## Referee Comment (RC1) · Anonymous Referee #1 · 1 Jun 2020

General comments:

This case study, apparently the first evaluation of how a parameterized wind farm affects the speed of a simulated gust front, is a useful addition to the literature. The figures and text are polished and generally free of mistakes. This is the start of a good paper, but it will benefit from substantial improvements to the scientific approach, analysis, and interpretation. The focus of my review is on major comments and substantive minor comments. A comprehensive list of technical corrections does not seem useful at this stage because much of the text is likely to change between this and future versions.

Specific comments:

1. One of my foremost concerns appears in the manuscript as a seemingly minor comment: "Such delays in the WFP outflow evolution could be artifacts of the 3-km model resolution" (lines 232-233). The same thought occurred to me, and it seems like a fundamental issue. A large fraction of the analysis in the manuscript is about the speeds of the simulated outflow with and without the wind farm's influence, and comparisons to observations. If the simulated speed's sensitivity to the computational grid is first order, then doesn't this call into question many of the manuscript's conclusions? At a minimum, other grid intervals should be tested to characterize this sensitivity. If the sensitivity is high, then the study needs to be redone based on simulations at a resolution for which results converge.

2. In the abstract, the stated goal of the manuscript is to "address the extent to which wind farms can modify thunderstorm outflow boundaries" (line 5). The actual analysis of results in the main body of the text is more modest than this statement suggests. The outflow's speed and some near-ground conditions are evaluated, but not its kinematic nor thermodynamic structures in the vertical. These also could exhibit important modifications, and the 4-D fields from the model, configured to provide high resolution in the lowest levels of the troposphere, should provide the opportunity to evaluate vertical structure.

3. The terms "propagate," "propagation," "propagating," etc. are used throughout the manuscript. However, I think the authors do not intend to refer to propagation velocity, but rather to velocity of the gust front relative to the ground. Velocity relative to the ground is the vector sum of two velocities: 1) velocity of the gust front's propagation through the air and 2) velocity of the air relative to the ground.

4. References to "resolution" in the manuscript should be corrected to "grid spacing," "grid interval," "model cell size," or something similar (there are many possible terms). It is an important distinction. The information that can actually be resolved with reasonable numerical accuracy is at a scale larger (coarser) than the grid interval. Specifically, the WRF Model's effective resolution is approximately 7x the grid interval (Skamarock, MWR, 2004). This means simulations on a grid with dx = 3 km have an effective resolution of approximately 21 km.

5. Lines 113-114 ("provides the initial visual of the outflow"): "Visual" is an adjective, not a noun. Also, I'm not sure what "initial visual" means. Is this just the initial (first) figure you are presenting in the manuscript?

6. Line 119 ("observation period of 5 minutes"): Does this mean that data were reported as a 5-minute average of samples taken every #? minutes or #? seconds? Is the reporting time centered on the 5-min period?

7. Line 121 ("located 5 km southwest of the wind farm"): The farm looks big. Is this distance of 5 km from the center of the farm, the southwestern edge, or to some other point?

8. Line 157 ("top-down view"): Consider a different adjective. This is an odd use of "top-down."

9. Lines 161-162 ("However, the outflow event in the simulation occurs too early"): The issue of timing deserves several sentences of additional explanation. Does the simulated moist convection occur too early overall, or does the timing seem about right except that the outflow is produced too early? Does the simulated outflow move too quickly at first?

10. Line 167 ("from the corresponding point"): Do you interpolate or choose the closest point?

11. Lines 172-173 ("possibly an artifact of the 5-min sampling in the observations as opposed to the 1-min sampling in the simulation"): Can you determine whether this is an artifact by averaging the model output in the same way the observations are averaged?

[Figure]

12. Lines 175-176 ("These model biases could be due to inaccuracies in the soil moisture that stem from differences in precipitation that occurred earlier in the day"): This is an interesting point. Can you check by running another simulation with higher soil moisture?

13. Lines 183-184 ("Regions of cooler temperatures (blue) indicate that the temperature in the WFP simulation is cooler than in the NWF simulation, suggesting faster movement of the outflow bringing cooler temperatures"): It might also be useful to show a plot of the difference in 10-m wind depicted as vectors.

14. Lines 185-186 ("Early in the outflow event, only subtle differences exist between the simulations upwind from the wind farm"): Do these differences surprise you? What do you think is causing them? Although it is true that the upwind differences are smaller than the largest differences on figs 5b,c where the gust front is most affected by the wind farm, the upwind differences are roughly the same as in many other parts of the domain in fig 5c. It would be helpful to reduce or completely eliminate the differences in simulations before the gust front nears the wind farm. Have you considered running the first part of the WFP simulation without the wind farm, writing a restart file at the output time just before the gust front reaches the farm, turning on WFP, then resuming the simulation by initializing with the restart file? I'm not sure if that is possible, but you might check.

15. Lines 191-192 ("The wind farm also appears to act as a barrier around which the outflow channels and accelerates, most notably immediately following passage over the wind farm"): Can you provide figures that support this interpretation? Can such channeling even be resolved at dx = 3 km? That equates to resolving features with a characteristic length scale of 21 km or so, which seems too coarse to depict channeling. (Please see the major comment about grid resolution.)

16. Line 199 ("suggesting the modified outflow in the wind farm case recovers after the initial disruption by the wind farm"): I am having trouble following this interpretation. The

analyisis of the data is not Lagrangian, it does not follow the gust front as it moves away from the wind farm and, presumably, is less influenced by it. Rather, the time series is at a point that does not move relative to the farm. That's why I don't know what to make of the term "recovery." Do you mean that changes in the moist convection and outflow with time at the observing station cause time series to appear more as they were before the gust front arrived?

17. Lines 210-212 ("wind speeds, the longer delay for the mixing ratio response indicates that the wind farm-modified outflow impacts absolute moisture more, suggesting that the change in moisture due to the outflow lags the change in temperature."): Maybe I am misreading, but isn't this a circular argument? Here is my interpretation: greater delay indicates greater impact, which suggests greater delay (i.e., lag). Also, I don't follow how you calculate what is a greater or lesser impact. The units of temperature and humidity are not the same. It's like trying to evaluate whether a flower is more colorful than a sprinter is fast. Would it clarify your point if you normalize changes in temperature and humidity by calculating standard deviations from sufficiently long temporal means? What is the physical explanation for why a greater lag can be interpreted to indicate a greater impact?

18. Lines 214-216 ("We measure the observed outflow propagation speed by tracking the reflectivity fine-line along a transect and recording its distance traveled every data update"): In the radar data, is the fine line at a consistent altitude at all times of your calculations? If not, then if the gust front is not vertical all the way to the top of the outflow (or at least to the highest altitude represented in the radar scans), wouldn't slope influence your calculations of location at fixed times and therefore your calculations of speed?

19. Lines 218-219 ("Both simulation and radar outflow are measured against a 5x5 km grid"): What does this mean?

20. Lines 220-221 ("Three separate measurement examinations are conducted for

each case"): That's an effective technique. Please explain this more. Three different people? The same person three different times? Something else?

21. Caption to figure 7 ("with the gradients in 10-m wind speed providing the position of the boundary to track"): Do you place the boundary where the gradient is highest? At the leading edge of the greatest change in the gradient? Somewhere else?

---

## Referee Comment (RC2) · Anonymous Referee #2 · 4 Jun 2020

Reviews corresponding to the article:" Observations and Simulations of a Wind Farm Modifying a Thunderstorm Outflow Boundary"

Authors: Jessica M. Tomaszewski, Julie K. Lundquist

General comment:

This manuscript investigates the impact of an onshore wind farm on the outflow boundary of a thunderstorm - a topic that has not been investigated so far. Hence, this manuscript is relevant and adds new knowledge to the wind energy community. Generally, the paper is well written and structured and the figures are well made! The only point that is unclear to the reviewer is the way they obtained the uncertainties (standard

deviations) shown in Fig. 7. Therefore, I ask the authors to add more information about this procedure (see specific comments, for more details).

Specific comments:

P11L220-221: "Three separate measurements examinations are conducted for each case...": could you be a bit more precise, as it is quite crucial to understand Fig. 7a). I don't understand how you actually obtained three different results: Did you conduct the measurements at three different points? Secondly, what did you do in case the transect intersected several times with the chosen threshold of 4 (m/s)/km?

---

## Author Comment (AC1) · 1 Sep 2020

**Tomaszewski et al. Responses to Reviewers**
**August 2020**

All reviewer comments appear in regular text below, while authors' responses appear in purple text. Line numbers referenced in the authors' responses refer to the revised document.

**Response to Anonymous Referee #1**

General comments:
This case study, apparently the first evaluation of how a parameterized wind farm affects the speed of a simulated gust front, is a useful addition to the literature. The figures and text are polished and generally free of mistakes. This is the start of a good paper, but it will benefit from substantial improvements to the scientific approach, analysis, and interpretation. The focus of my review is on major comments and substantive minor comments. A comprehensive list of technical corrections does not seem useful at this stage because much of the text is likely to change between this and future versions.

Thank you for your thoughtful review and helpful comments.

Specific comments:
1. One of my foremost concerns appears in the manuscript as a seemingly minor comment: "Such delays in the WFP outflow evolution could be artifacts of the 3-km model resolution" (lines 232-233). The same thought occurred to me, and it seems like a fundamental issue. A large fraction of the analysis in the manuscript is about the speeds of the simulated outflow with and without the wind farm's influence, and comparisons to observations. If the simulated speed's sensitivity to the computational grid is first order, then doesn't this call into question many of the manuscript's conclusions? At a minimum, other grid intervals should be tested to characterize this sensitivity. If the sensitivity is high, then the study needs to be redone based on simulations at a resolution for which results converge.

Thank you for your thoughtful question. We have considered the effect of horizontal grid resolution at length. In one of our previous papers, we compared 1-km, 3-km and 9-km horizontal grid spacing simulations of wind plant wakes (Tomaszewski and Lundquist, 2020, *Geoscientific Model Development*), and generally found that use of a 3-km horizontal grid tended to result in similar wake impacts as use of a finer 1-km horizontal grid. Larger deviations tended to occur when coarser resolution (on the order of 9-km grid) was used. While we do not have the computational resources to run this case at 1-km horizontal grid resolution, we can coarsen the resolution, and so we have re-run this scenario at 4.5-km grid resolution. The subtle difference (see figures below) between the 3-km (colored) and the 4.5-km (grey) suggests that the delay in the WFP outflow evolution is due to initial and boundary conditions rather than the computational grid, which we have also added to the text in line 242:

"Such delays in the WFP outflow evolution could be artifacts of the 3-km grid resolution or more likely the initial and boundary conditions"

[Figure]

2. In the abstract, the stated goal of the manuscript is to "address the extent to which wind farms can modify thunderstorm outflow boundaries" (line 5). The actual analysis of results in the main body of the text is more modest than this statement suggests. The outflow's speed and some near-ground conditions are evaluated, but not its kinematic nor thermodynamic structures in the vertical. These also could exhibit important modifications, and the 4-D fields from the model, configured to provide high resolution in the lowest levels of the troposphere, should provide the opportunity to evaluate vertical structure.

We have modified the abstract to more accurately reflect the goals of the manuscript in line 5:

"We use the Weather Research and Forecasting model and its Wind Farm Parameterization to address the extent to which wind farms can modify the near-surface environment of thunderstorm outflow boundaries."

Because vertical structure is interesting, we also included the below figure (new Fig. 6) to give insight on the structures in the vertical impacted by the wind farm, with additional text in lines 197-198:

"A vertical cross-section of the temperature difference between the simulations taken at 23:22 (dashed line in Fig. 5b) shows that the wind farm (black X) impacts the outflow from the surface to ~2 km (Fig. 6)."

[Figure]

Vertical cross-section of the temperature difference between the Wind Farm Parameterization (WFP) and no wind farm (NWF) simulations immediately following outflow boundary passage over the wind farm, corresponding to the dashed line in (Fig. 5b). Note the y-axis ticks are not spaced linearly due to the increasingly coarse vertical grid spacing at higher model levels.

3. The terms "propagate," "propagation," "propagating," etc. are used throughout the manuscript. However, I think the authors do not intend to refer to propagation velocity, but rather to velocity of the gust front relative to the ground. Velocity relative to the ground is the vector sum of two velocities: 1) velocity of the gust front's propagation through the air and 2) velocity of the air relative to the ground.

We have replaced all occurrences of "propagate" with more appropriate words, e.g.,

replaced "propagation speed" with "ground-relative velocity" in lines 3, 11, 216, 221, 244;

replaced "outflow propagation speed" with "outflow speed" in lines 7, 217, 222, 229, 282, 288;

replaced "propagation speed" with "speed" in lines 86, 195, 217, 230.

4. References to "resolution" in the manuscript should be corrected to "grid spacing," "grid interval," "model cell size," or something similar (there are many possible terms). It is an important distinction. The information that can actually be resolved with reasonable numerical accuracy is at a scale larger (coarser) than the grid interval. Specifically, the WRF Model's effective resolution is approximately 7x the grid interval (Skamarock, MWR, 2004). This means simulations on a grid with dx = 3 km have an effective resolution of approximately 21 km.

We agree with the distinction and have replaced all occurrences of "resolution" with "grid spacing" or similar, or ensured that comments about "resolution" specify "grid resolution" to distinguish them from effective model resolution.

5. Lines 113-114 ("provides the initial visual of the outflow"): "Visual" is an adjective, not a noun. Also, I'm not sure what "initial visual" means. Is this just the initial (first) figure you are presenting in the manuscript?

Thank you for catching our typo. We have replaced "visual" with "visualization" and clarified that the radar provides this study with an initial view of the outflow in line 114:

"…provides the initial visualization for this study of the outflow."

6. Line 119 ("observation period of 5 minutes"): Does this mean that data were reported as a 5-minute average of samples taken every #? minutes or #? seconds? Is the reporting time centered on the 5-min period?

We provided more information on the observations in line 120:

"Sampling intervals vary from 3 to 60 s depending on the sensor, and data are reported as 5-minute averages centered on the 5-minute period."

7. Line 121 ("located 5 km southwest of the wind farm"): The farm looks big. Is this distance of 5 km from the center of the farm, the southwestern edge, or to some other point?

We have clarified on line 122:

"The Abernathy surface station is located 5 km southwest of the southwest corner of the wind farm."

8. Line 157 ("top-down view"): Consider a different adjective. This is an odd use of "top-down."

"Top-down view" has been replaced with "plan view" in line 158.

9. Lines 161-162 ("However, the outflow event in the simulation occurs too early"): The issue of timing deserves several sentences of additional explanation. Does the simulated moist convection occur too early overall, or does the timing seem about right except that the outflow is produced too early? Does the simulated outflow move too quickly at first?

We have clarified the text in line 163 to read:

"However, the simulated moist convection and subsequent outflow boundary occurs too early."

10. Line 167 ("from the corresponding point"): Do you interpolate or choose the closest point?

We have clarified in line 167 that we use the closest point:

"we plot a time series from the nearby Abernathy West Texas Mesonet surface station against that from the corresponding closest point in the model domain (grey triangle in Fig. 5)."

11. Lines 172-173 ("possibly an artifact of the 5-min sampling in the observations as opposed to the 1-min sampling in the simulation"): Can you determine whether this is an artifact by averaging the model output in the same way the observations are averaged?

Plots averaging the model output over 5-min periods to match the observations reveal a similar delay in wind speed decrease, confirming the averaging is the likely reason for the differences in the time series in Fig. 4a. We updated the text in lines 174-175 to read:

"…likely an artifact of the 5-min sampling in the observations as opposed to the 1-min sampling in the simulation, verified by plotting a 5-minute average of the simulation results (dashed line in Fig. 4a)."

[Figure]

12. Lines 175-176 ("These model biases could be due to inaccuracies in the soil moisture that stem from differences in precipitation that occurred earlier in the day"): This is an interesting point. Can you check by running another simulation with higher soil moisture?

Unfortunately, we have only extremely limited remaining computational resources, so a new soil moisture simulation is not possible. But it is a great suggestion, so we incorporate this idea in the conclusions for a subsequent investigation in line 295:

"This case study could motivate a larger-scale climatology of additional outflow-wind-farm interaction events, including different environments with variable soil moistures or other meteorological properties."

13. Lines 183-184 ("Regions of cooler temperatures (blue) indicate that the temperature in the WFP simulation is cooler than in the NWF simulation, suggesting faster movement of the outflow bringing cooler temperatures"): It might also be useful to show a plot of the difference in 10-m wind depicted as vectors.

We have chosen to display the difference in the 10-m wind vectors in Fig. 5 as two sets of wind barbs, one from each simulation, but we've added a statement to the text specifically acknowledging this component of the figure in lines 186-188:

"Indeed, the dark wind barbs in Fig. 5 representing winds from the WFP simulation indicate stronger winds present (by 5-10 kts) in cooler (blue) regions than in the NWF simulation (light wind barbs)."

14. Lines 185-186 ("Early in the outflow event, only subtle differences exist between the simulations upwind from the wind farm"): Do these differences surprise you? What do you think is causing them? Although it is true that the upwind differences are smaller than the largest differences on figs 5b,c where the gust front is most affected by the wind farm, the upwind differences are roughly the same as in many other parts of the domain in fig 5c. It would be helpful to reduce or completely eliminate the differences in simulations before the gust front nears the wind farm. Have you considered running the first part of the WFP simulation without the wind farm, writing a restart file at the output time just before the gust front reaches the farm, turning on WFP, then resuming the simulation by initializing with the restart file? I'm not sure if that is possible, but you might check.

Differences in the wind farm and no wind farm simulations due to the presence of the wind farm parameterization are an ongoing area of research. These upwind differences may arise from the generation of gravity wave (e.g., Smith, 2009; Allaerts and Meyers, 2018, 2019). We have taken the reviewer's suggestion and tried using a non-WFP restart file to initialize a WFP simulation as the reviewer suggested, but because some of the required output fields for the WFP simulation (i.e. POWER) are not included in the non-WFP restart files, this is not possible.

We have included a comment in the text about the upwind differences possibly being due to gravity waves, and included the references given above in lines 190-191:

"Early in the outflow event, only subtle differences exist between the simulations upwind from the wind farm (Fig. 5a), likely arising from the generation of gravity waves (Smith, 2009; Allaerts and Meyers, 2018, 2019)."

15. Lines 191-192 ("The wind farm also appears to act as a barrier around which the outflow channels and accelerates, most notably immediately following passage over the wind farm"): Can you provide figures that support this interpretation? Can such channeling even be resolved at dx = 3 km? That equates to resolving features with a characteristic length scale of 21 km or so, which seems too coarse to depict channeling. (Please see the major comment about grid resolution.)

The reviewer raises a good point that fine channeling will not be fully resolved. We should not have used the word "channeling", as we intend to suggest that the cooler regions emerging on either side of the wind farm in Fig. 5 are effects of the outflow boundary being redirected around the wind farm. We do understand the concern with this wording and have updated the text in line 196-197 to reflect our hypothesis better:

"The cooler regions of faster moving air emerging on both sides of the wind farm during outflow passage suggests flow is being redirected around the wind farm (Fig. 5b)."

16. Line 199 ("suggesting the modified outflow in the wind farm case recovers after the initial disruption by the wind farm"): I am having trouble following this interpretation. The analyisis of the data is not Lagrangian, it does not follow the gust front as it moves away from the wind farm and, presumably, is less influenced by it. Rather, the time series is at a point that does not move relative to the farm. That's why I don't know what to make of the term "recovery." Do you mean that changes in the moist convection and outflow with time at the observing station cause time series to appear more as they were before the gust front arrived?

We agree that this message needs clarification and have amended this sentence in lines 204-206:

"A secondary pulse of increased wind speeds occurs in both simulation cases and reaches similar magnitudes, suggesting the modified outflow in the wind farm case does not experience notable changes after the initial disruption by the wind farm (Fig. 7a)."

17. Lines 210-212 ("wind speeds, the longer delay for the mixing ratio response indicates that the wind farm-modified outflow impacts absolute moisture more, suggesting that the change in moisture due to the outflow lags the change in temperature."): Maybe I am misreading, but isn't this a circular argument? Here is my interpretation: greater delay indicates greater impact, which suggests greater delay (i.e., lag). Also, I don't follow how you calculate what is a greater or lesser impact. The units of temperature and humidity are not the same. It's like trying to evaluate whether a flower is more colorful than a sprinter is fast. Would it clarify your point if you normalize changes in temperature and humidity by calculating standard deviations from sufficiently long temporal means? What is the physical explanation for why a greater lag can be interpreted to indicate a greater impact?

We agree that comparing the extent of impact on temperature and humidity doesn't make sense and is not of much value to the purpose of this study. We have removed discussion comparing the impacts between these variables.

18. Lines 214-216 ("We measure the observed outflow propagation speed by tracking the reflectivity fine-line along a transect and recording its distance traveled every data update"): In the radar data, is the fine line at a consistent altitude at all times of your calculations? If not, then if the gust front is not vertical all the way to the top of the outflow (or at least to the highest altitude represented in the radar scans), wouldn't slope influence your calculations of location at fixed times and therefore your calculations of speed?

Thank you for this helpful comment about the effective altitude of the radar measurements. We have reassessed the radar location and the geometry of the relevant measurements. The radar is southwest (220°) of the portion of the outflow boundary of interest, which is moving nearly to the southeast at a heading of (120°) (Fig. 1). We are relieved to note that the feature therefore maintains an approximately constant distance to the radar even as the feature moves. Therefore, the feature maintains a similar height during the brief period of interest. We

have added the following text to the manuscript to point out this important consideration in lines 218-221:

"As the radar is southwest (220°) of the portion of the outflow boundary of interest, which is moving nearly to the southeast at a heading of (120°) (Fig. 1), we note that the feature therefore maintains an approximately constant distance to the radar and thus height above ground even as the feature moves, thus not impacting our calculations of ground-relative outflow speed."

19. Lines 218-219 ("Both simulation and radar outflow are measured against a 5x5 km grid"): What does this mean?

We used the 5x5 km grid shown in Fig. 8 as a visual guideline to aid in tracking. We updated Fig. 8's caption to say:

"....The schematic in (b) shows the process for calculating the simulation outflow speed, where the line through the domain shows the transect along which speed was measured, with the largest gradients in 10-m wind speed providing the position of the boundary to track, and the underlying 5-km-by-5-km grid providing a visual guideline to aid in tracking."

20. Lines 220-221 ("Three separate measurement examinations are conducted for each case"): That's an effective technique. Please explain this more. Three different people? The same person three different times? Something else?

We agree that clarification is needed here. We have added to the text in lines 226-230:

"Three separate measurements are conducted for each case (i.e., radar, WFP simulation, and NWF simulation) to account for human error. Each examination is conducted three times the same way by the same person over the same transect to generate multiple estimates of outflow speed. Conducting this qualitative measurement is a dynamic process, and once a point on the 4 m/s/km transect was chosen, the measurement tracker tried to follow that same point, even if the transect intersected the 4 m/s/km contour in multiple places. The averages of each case are plotted in Fig 8a...."

21. Caption to figure 7 ("with the gradients in 10-m wind speed providing the position of the boundary to track"): Do you place the boundary where the gradient is highest? At the leading edge of the greatest change in the gradient? Somewhere else?

We placed the boundary where the gradient is largest, which we clarified in Fig. 8's caption:

"...with the largest gradients in 10-m wind speed providing the position of the boundary to track."

**Response to Anonymous Referee #2**

General comment:
This manuscript investigates the impact of an onshore wind farm on the outflow boundary of a thunderstorm - a topic that has not been investigated so far. Hence, this manuscript is relevant and adds new knowledge to the wind energy community. Generally, the paper is well written and structured and the figures are well made!

Thank you for your thoughtful review and helpful comments.

The only point that is unclear to the reviewer is the way they obtained the uncertainties (standard C1 WESD Interactive comment Printer-friendly version Discussion paper deviations) shown in Fig. 7. Therefore, I ask the authors to add more information about this procedure (see specific comments, for more details).

Specific comments: P11L220-221: "Three separate measurements examinations are conducted for each case. . ."; could you be a bit more precise, as it is quite crucial to understand Fig. 7a). I don't understand how you actually obtained three different results: Did you conduct the measurements at three different points? Secondly, what did you do in case the transect intersected several times with the chosen threshold of 4 (m/s)/km?

We agree that clarification is needed here. We have added to the text in lines 226-230:

"Three separate measurements are conducted for each case (i.e., radar, WFP simulation, and NWF simulation) to account for human error. Each examination is conducted three times the same way by the same person over the same transect to generate multiple estimates of outflow speed. Conducting this qualitative measurement is a dynamic process, and once a point on the 4 m/s/km transect was chosen, the measurement tracker tried to follow that same point, even if the transect intersected the 4 m/s/km contour in multiple places. The averages of each case are plotted in Fig 8a...."

---

## Author Response (AR3)

**Tomaszewski et al. Responses to Reviewers**
**October 2020**

All reviewer comments appear in regular text below, while authors' responses appear in purple text. Line numbers referenced in the authors' responses refer to the revised document.

**Response to Anonymous Referee #1**

The authors' replies to the reviewers and revisions to the manuscript are excellent. Only a few very minor points need to be reviewed before publishing.

Thank you for your thoughtful review and helpful comments.

Line 93: Confirm that "wind farm-near environment interactions" is the correct wording.

We have changed line 93 to read: "wind farm near-environment interactions".

Line 128: "comprising" should be "composing"

Line 128 now says "composing".

Line 268: Consider "wind energy infrastructure" rather than "wind energy," or use a noun other than "deployment" -- "harvesting" maybe

We have updated the phrase in line 268 to the more complete "wind energy infrastructure".

Throughout: Some references to direction have a "-ward" suffix (e.g., "move southeastward" in line 105) but some do not (e.g., "propagated southeast" in line 160). The forms of such terms should be made consistent, and "-ward" is the better option, indicating toward a stated direction.

All references to direction now include the "-ward" suffix (see lines 2, 85, and 160).

[revised manuscript text omitted]